# Early Growth and Development and Nonlinear Model Fitting Analysis of Ashidan Yak

**DOI:** 10.3390/ani13091545

**Published:** 2023-05-05

**Authors:** Guangyao Meng, Yongfu La, Qi Bao, Xiaoyun Wu, Xiaoming Ma, Chun Huang, Min Chu, Chunnian Liang, Ping Yan

**Affiliations:** 1Lanzhou Institute of Husbandry and Pharmaceutical Sciences, Chinese Academy of Agricultural Sciences, Lanzhou 730050, China; 2Key Laboratory of Animal Genetics and Breeding on Tibetan Plateau, Ministry of Agriculture and Rural Affairs, Lanzhou 730050, China

**Keywords:** Ashidan yak, growth and development, growth curve, growth traits

## Abstract

**Simple Summary:**

Ashidan yak is a new breed of hornless yak developed by Chinese scientists, which has an important economic value. However, little is known about the growth of Ashidan yaks. This study analyzed the body weight and body size measurements of 260 female Ashdan yaks and compared the performance of five nonlinear models (Logistic model, Gompertz model, Brody model, von Bertalanffy model and Richards model). Our results showed that the early growth and development of Ashidan yak change with the seasons, and the Richards model performs better among the five models.

**Abstract:**

Understanding animal growth plays an important role in improving animal genetics and breeding. In order to explore the early growth and development law of Ashidan yak, the body weight (BW), wither height (WH), body oblique length (BL) and chest girth (CG) of 260 female Ashidan yaks were measured. These individuals grew under grazing conditions, and growth traits were measured at 6, 12, 18 and 30 months of age. Then the absolute growth and relative growth of Ashidan yak were calculated, and five nonlinear models (Logistic model, Gompertz model, Brody model, von Bertalanffy model and Richards model) were used to fit the growth curve of Ashidan yak. The fitting effect of the model was evaluated according to MSE, AIC and BIC. The results showed that the growth rate of Ashidan yak was the fastest from 12 to 18 months old, and the growth was slow or even stagnant from 6 to 12 months old. The AIC and BIC values of the Richards model were the lowest among the five models, with an AIC value of 4543.98 and a BIC value of 4563.19. The Richards model estimated body weight at 155.642 kg. In summary, the growth rate of female Ashidan yak changes with the seasons, growing faster in warm seasons and slower in cold seasons. Richards model is the best model to describe the growth curve of female Ashidan yak in five nonlinear models.

## 1. Introduction

Yak (*Bos grunniens*) is the main livestock living in the Qinghai Tibet Plateau and its surrounding areas [1]. It is an advantageous animal species that can make full use of grassland resources and carry out animal production. Additionally, it is a unique genetic resource in plateau areas. It can provide meat, milk, wool and fuel for people in pastoral areas. It is closely related to the life of people in plateau areas [2,3,4,5]. Ashidan yak is a new breed of hornless yak bred by Chinese scientists using the population subculture breeding method. (Figure 1). Ashdan yak is based on the Qinghai Plateau polled yak as the parent, the application of testcross and controlled inbreeding, combined with the use of molecular maker-assisted breeding method to shorten the breeding cycle after four generations of breeding. It has the characteristics of good meat production performance, strong stress resistance, high reproductive performance, easy enclosure feeding and high economic benefits [6]. At present, studies on Ashidan yak include association analysis between copy number variants (CNV) and growth traits and genome-wide prediction [6,7,8,9].

Understanding growth is the basis of many biological fields and is also crucial for industries based on animal and plant production [10]. At present, in the research of livestock and poultry, the regulation of the growth and development of animals is usually reflected by fitting the growth curve, which can reflect the relationship between individual development speed and maturity rate [10,11]. The research on the growth curve has a long history. Nonlinear models have been widely used in the growth simulation of animals and plants because they can explain the growth pattern in the process of the growth cycle [12]. Frequently used nonlinear models include cumulative growth, the Logistic model [13], the Gompertz model [14], the Brody model [15], the Richards model [16], the von Bertalanffy model [17], etc. Studies have shown that the parameters derived from the model can be inherited and respond to the breeding program [18,19]. Thus far, a large number of researchers have used animal weight data for growth curve fitting in order to improve breeding efficiency and promote breeding work. For example, the growth curve models of different breeds of beef cattle, such as Brahman cattle, Angus cattle and Polled Hereford cattle, have been established, which can effectively understand the growth and development process of different breeds of beef cattle [20,21,22]. However, due to the impact of the harsh environment in the plateau area, the fitting research of growth curve of yak had been a largely under explored domain. The main purpose of this study was to explore the early growth and development of Ashidan yak measured at four time points and evaluate the performance of five growth curve models.

## 2. Materials and Methods

### 2.1. Ethics Statement

The research scheme was approved by the Key Laboratory of Yak Breeding Engineering of Gansu Province, Lanzhou Institute of Husbandry and Pharmaceutical Sciences (No.LIHPS-CAAS-2017-115). The body measurements were performed in strict accordance with the Guide for the Care and USE of Laboratory Animals [23].

### 2.2. Sample Collection and Growth Traits

The experimental population used in this study is 260 healthy female Ashidan yaks. They live at an altitude of more than 3200 m, raised by grazing all year round and supplemented with feed in cold seasons. The feeding standards and growth environment are basically consistent. The body weight and body size data of these individuals at the age of 6, 12, 18 and 30 months were tracked and measured, and descriptive statistics were carried out using Excel software.

### 2.3. Fitting of Nonlinear Models 

The logistic model, Gompertz model, Brody model, von Bertalanffy model and Richards model were used to fit the growth curve of early weight data of Ashidan yak (Table 1). The degree of fitting of the model to the data was evaluated according to the mean square errors (MSE), Akaike’s information criterion (AIC) and Bayesian information criterion (BIC). They were defined as:$$AIC=n*\ln\left(\frac{SSE}{n}\right)+2k,$$
$$BIC=n*\ln\left(\frac{SSE}{n}\right)+k*\ln\left(n\right)$$
where *k* is the number of parameters in the model and n is the sample size. 

### 2.4. Calculation of Growth and Development Indicators

The absolute growth rate (G) and relative growth rate (R) were calculated to reflect the growth and development of Ashidan yak. The calculation formula is as follows:$$G=\left(W_{1}-W_{0}\right)/\left(t_{1}-t_{0}\right)$$
$$R=\frac{W_{1}-W_{0}}{\left[\frac{W_{1}+W_{0}}{2}\right]}\times 100\%$$
where *W*_1_ is the weight (body size) at the end of the experimental phase, *W*_0_ is the weight (body size) at the beginning of the experimental phase, *t_1_* is the age at the end of the experimental phase, and *t_0_* is the age at the beginning of the experimental phase.

### 2.5. Statistical Analysis

The feeding mode of Ashidan yak was grazing all year round. Due to the influence of the particular climate in the plateau area, it was difficult to track and determine the growth characteristics of Ashidan yak, so the number of samples gradually decreased with the increase in time. Detailed data information is shown in Table 2. 

The EXCEL software calculated two growth indicators, absolute and relative growth and draw curves. The nonlinear regression analysis in the regression analysis program of SPSS 16.0 software (IBM, New York, NY, USA) was used to obtain the optimal estimation of model equation parameters. According to the weight record data of different months of age, the optimal estimation values of each model parameter (A, B, K and m) were fitted and calculated, and the growth model was established to calculate the inflection point weight, inflection point age and growth model index.

## 3. Results

### 3.1. Statistics of Growth Traits

The descriptive statistical information of different growth traits of Ashidan yak is shown in Table 2. The coefficient of variation (CV) of body weight (BW) of Ashidan yak is 9.75–12.16%, and the CV of body size is 4.17–7.86%. The CV of body weight is larger than that of body size. The maximum and minimum values of growth traits of Ashidan yak at the same time are quite different, the degree of dispersion is high, and the values of standard deviation (SD) and CV are also large, which indicates that there were phenotypic variations and differences in growth and development of Ashidan yak. 

In the early growth and development process, the body weight of Ashidan yak showed a fluctuating trend of decreasing and increasing (Figure 2a). The body weight of Ashidan yak slightly decreased from 6 to 12 months of age but increased from 12 to 18 months of age and 18 to 30 months of age. Figure 2b,c show the curves of absolute and relative growth calculated and drawn based on the measured body weight data. The results showed that the absolute and relative growth of body weight of Ashidan yak were at a negative growth level from 6 to 12 months of age, at a positive growth level from 12 to 18 months of age and at a positive growth level from 18 to 30 months of age, but lower than that of 12–18 months of age. Ashidan yak was in the growth stagnation period at 6–12 months of age, and the growth rate and value of body weight were in the weak negative growth state. Ashidan yak was in the growth peak period at 12–18 months of age, and the growth rate was relatively large. Ashidan yak was in the growth period at 18–30 months of age, but the growth rate decreased.

### 3.2. Cumulative Growth, Relative Growth and Absolute Growth of Wither Height of Ashidan Yak

During the early growth and development of Ashidan yak, the wither height showed a trend of increasing (Figure 3a). The wither height of Ashidan yak increased at 6–12 months of age, 12–18 months of age and 18–30 months of age. Figure 3b,c show the curves of absolute and relative growth calculated and drawn based on the measured wither height data. The results showed that the absolute and relative growth of wither height of Ashidan yak were at a positive growth level during early growth and development. Among them, the wither height of some individuals increased rapidly during the period of 12–18 months of age, while the average wither height of the Ashidan yak population was basically the same during 6–12 months of age and 12–18 months of age, and the growth rate decreased during 18–30 months of age.

### 3.3. Cumulative Growth, Relative Growth and Absolute Growth of Body Oblique Length of Ashidan Yak

In the early growth and development process of Ashidan yak, the average body oblique length showed an increasing trend, while the maximum value showed a fluctuation trend (Figure 4a). Body oblique length refers to the straight line distance from the most front end of the humerus to the back end of the ischial tuberosity. The measurement tool for the body oblique length of the yak is a soft ruler, which is used close to the trunk of the yak. Additionally, the yak undergoes a long cold season, their trunk changes, some individuals become thinner, and the body oblique length may also become smaller. The maximum body oblique length of Ashidan yak decreased from 6 to 12 months of age, but the average and minimum of Ashidan yak increased from 12 to 18 months of age and from 18 to 30 months of age. Figure 4b,c show the curves of absolute and relative growth calculated and drawn based on the measured body oblique length data. From 6 to 12 months of age, the average growth rate and growth value of body oblique length of Ashidan yak were basically in a positive growth state. The growth rate of the body oblique length of Ashidan yak at 12 to 18 months of age increased, and the growth rate of body length at 18 to 30 months of age increased again, with the fastest growth rate.

### 3.4. Cumulative Growth, Relative Growth and Absolute Growth of Chest Girth of Ashidan Yak

During the early growth and development of Ashidan yak, the average chest circumference and the maximum chest circumference showed a trend of decreasing and increasing, while the minimum chest circumference showed an increasing trend (Figure 5a). The average chest circumference of Ashidan yak decreased from 6 to 12 months of age but increased from 12 to 18 months of age and 18 to 30 months of age. Figure 5b,c show the curves of absolute and relative growth calculated and drawn based on the measured chest circumference data. The results showed that the absolute growth and relative growth of breast circumference of Ashidan yak were at negative growth levels from 6 to 12 months of age, positive growth levels from 12 to 18 months of age, and positive growth levels from 18 to 30 months of age. From 6 to 12 months of age, the growth rate and growth value of the chest circumference of Ashidan yak were in a weak negative growth state. The chest circumference of Ashidan yak increased rapidly at 12 to 18 months of age but decreased significantly at 18 to 30 months of age, and the chest circumference of Ashidan yak increased the fastest at 12 to 18 months of age.

### 3.5. Results of Nonlinear Model Fitting for Growth Traits of Ashidan Yak

The estimated growth parameters are shown in Table 3. Estimated body weights (A) ranged from 155.462 kg (Richards) to 6527.970 kg (Brody) for Ashidan yaks. Estimated wither heights ranged from 99.848 cm (Richards) to 105.395 cm (Brody). Notably, the estimated body length and chest girth were so large that they did not match the actual value. Table 4 shows the goodness of different models for body weight. According to the criteria of MSE, AIC and BIC, the Richards model was the best model, and its MSE, AIC and BIC values were the lowest. The Brody model was the worst since it had the highest value.

### 3.6. Growth Curve of Body Weight of Ashidan Yak

The fitting results of five nonlinear models to the body weight of Ashidan yak are shown in Figure 6. The theoretical values of the growth and development stages of Ashidan yak from 6 months to 30 months showed a steady upward trend in the five models. Among the five models, the fitted curve of body weight of Ashidan yak by the Brody model was significantly different from the measured growth curve, while the fitted curve of body weight by the Logistic model, Gompertz model, von Bertalanffy model and Richards model were close to the actual growth curve, and the three curves were consistent. In the figure, three kinds of curves overlap. The logistic model, Gompertz model, von Bertalanffy model and Richards model showed that the estimated weight at 12 months of age was higher than the measured value, the estimated weight at 6 months and 18 months of age was lower than the measured value, and the estimated weight at 30 months of age was close to the measured value.

## 4. Discussion

### 4.1. Early Growth and Development Law of Ashidan Yak

Ashidan yak is famous for its hornless and high meat production performance. It became one of China’s new national breeds of livestock in 2019. To the best of our knowledge, there is no research on the growth and development of hornless yak. The yak population used in this study grows in the plateau area with an altitude of 2900–4600 m. This place has no four seasons; the whole year is divided into long, cold seasons and short, warm seasons. It belongs to the continental climate of warm, cool and semi-humid plateau. The Ashidan yak production management is grazing all year round and free feeding. This study analyzed the body weight and body size data of Ashidan yak in the early stage of growth and development and found that there were varying degrees of variation in body weight and body size, and the degree of dispersion was high, indicating that there were great differences in the growth and development of Ashidan yak. In the actual production work, the growth and development of yaks are affected not only by genetic factors but also by the special environment in the plateau area. Therefore, when studying the growth and development of Ashidan yak, we should comprehensively consider the factors affecting its growth and development.

Body weight and body size are important indicators to reflect the growth and development of livestock and can measure their production performance. In this study, the growth and development indicators of Ashidan yak were calculated, and the growth curves were drawn based on the recorded data of body weight, body height, body oblique length and chest girth. It was found that the early growth and development of Ashidan yak showed a fluctuation trend of decreasing and increasing, and there was a negative growth phenomenon of body weight and chest circumference during 6–12 months of age. The causes of this phenomenon are determined by the characteristics of yak development and environmental factors. At present, the feeding mode of yak is mainly grazing, and its growth and development have obvious seasonality. However, Ashidan yak experiences the cold season wither period during 6–12 months of age, and the quality of herbage decreases, which leads to its failure to obtain adequate nutrition. Cold and forage shortages affected the growth status and growth rate of Ashidan yak. The growth and development patterns of different breeds of animals are specific [24]. Previous studies on the growth and development of yaks have found that the growth and development of yaks have obvious seasonality [25], and the results of this study are thus consistent with these findings.

### 4.2. Fitting Analysis of Nonlinear Model

The main purpose of fitting the growth curve is descriptive and predictive [26]. Many factors affect the performance of the growth curve model, such as sample size, data structure (time interval between records) and environment [27]. Among the five nonlinear models applied in this study, the Logistic and Gompertz models are characterized by a fixed inflection point. The growth rate before the inflection point is faster, the growth rate after the inflection point is slower, and the overall curve is “S” shaped [28,29]. Brody’s model is characterized by a gradual growth process, while von Bertalanffy’s model is an “S-shaped” growth curve model with variable inflection points [30]. Due to the adaptability of Ashidan yak to the low oxygen environment in the plateau and the influence of the cold season environment, the body size characteristics of Ashidan yak have the phenomenon of high and low in the growth and development process, resulting in the low fitting degree of the five nonlinear models for the body size characters of Ashidan yak. In this study, it was found that the Richards model had the best fit for the body weight of Ashidan yak. The Richards model is a sigmoid function with more parameters, and its simulated “S” curve is more flexible [27]. The specific parameter (m) of the Richards model can evaluate the shape of the growth curve systematically and concisely, so that its fitting effect surpasses other models [31]. On the other hand, previous studies have shown that the Gompertz model has the best performance in fitting the growth curve of Chinese beef cattle [30], which is inconsistent with the results of this study. Notably, the experimental group in their study was Simmental beef cattle compared to yaks in our study. In addition, the experimental data have a decisive significance for the fitting degree of the model. Compared with our study, they used a larger number of samples and a shorter interval of weight data, and these may lead to better model performance. In our study, the body weight and body size of yaks were measured manually at an altitude of more than 3200 m, which requires overcoming the difficulties caused by altitude sickness and cold environments. Therefore, we used a measuring interval of 6 months and 12 months. In a previous study on a similar population, the researchers constructed the growth curve of Maiwa yak with body weight data and found that the curve deviated from the “S” type [25]. This study also found that the growth curve of Ashidan yak deviated from the “S” type. After the birth of the yak, the environment mainly affects the growth and development of the yak. The warm season is warm, the pasture is lush, and yak grow faster. If there is a heavy snow disaster in the cold season, the growth of yaks is extremely slow. This is the main reason for the deviation of yak growth curve from “S” type.

In this experiment, the estimated weight of the Logistic, Gompertz and von Bertalanffy models at 6 months of age was lower than the measured values, which may be related to the lactation stage and environmental characteristics. From birth to 6 months of age, the pasture area is in the warm season, the pasture is fertile, the calf can obtain sufficient milk from the mother, and the growth is fast. At 12 months of age, the estimated value of the model was higher than the measured value. At 18 months of age, the estimated value of the model was lower than the measured value, which may be influenced by the environment, resulting in a change in the growth and development rate of Ashidan yak. In another study, the researchers measured carcass weight and other traits of Brahman and Angus hybrid cattle after feeding in cold and warm seasons. It was found that the carcass weight of the warm season was significantly higher than that of the cold season [32]. Therefore, for the growth curve analysis, it is essential to understand the influence of different factors on growth and the duration of influence [33]. Furthermore, estimation of the heritability of growth curve parameters will be helpful for genome selection of body weight and body size based on growth curve parameters in the future [27]. In this study, the Richards model was recommended to model the growth of Ashidan yak, but it may need to be validated in other populations with shorter measurement intervals before implementing in breed improvement programs.

## 5. Conclusions

According to relative growth and absolute growth, this study found that the early growth and development of Ashidan yak had obvious seasonality, and based on the comparison of AIC and BIC, the Richards model has the best fitting effect among the target five growth curve models. Female Ashidan yaks follow the growth and development pattern of yaks. They grow fast in the warm season, grow slowly in the cold season, and are greatly affected by the environment, so supplementary feeding of Ashidan yaks should be increased in the cold season. From this research on the nonlinear model describing the growth curve of female Ashidan yak, the applicability and defects of the models to female Ashidan yak can be obtained, which will help the subsequent genetic research based on growth parameters.

## Figures and Tables

**Figure 1 animals-13-01545-f001:**
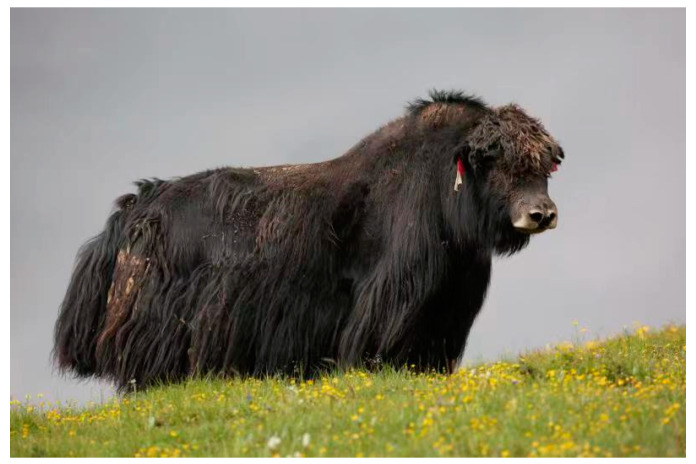
Ashidan yak.

**Figure 2 animals-13-01545-f002:**
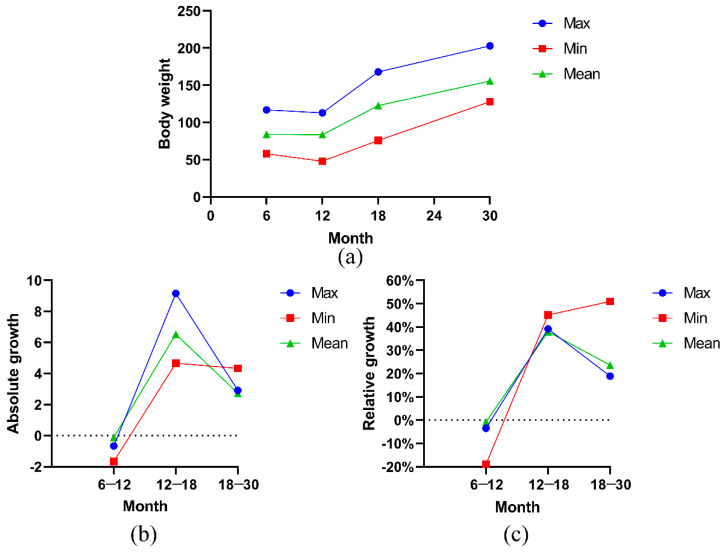
(**a**) Cumulative growth curve of body weight (BW); (**b**) absolute growth curve of BW; (**c**) relative growth curve of BW.

**Figure 3 animals-13-01545-f003:**
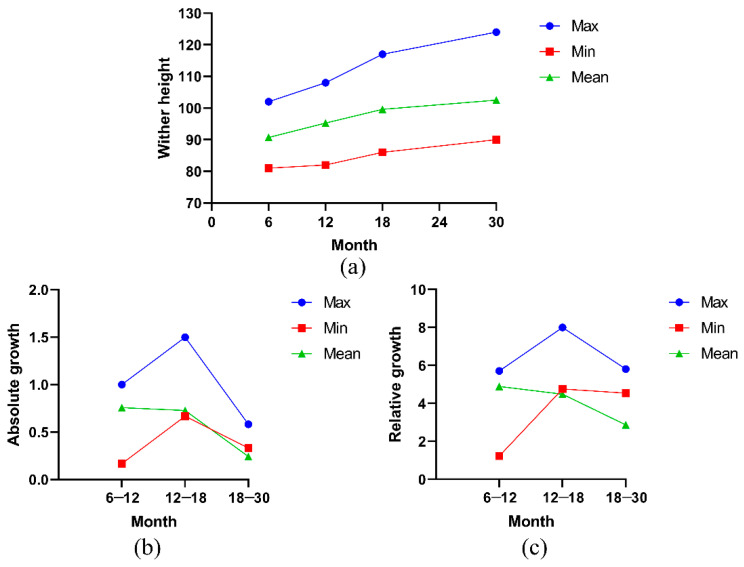
(**a**) Cumulative growth curve of wither height (WH); (**b**) absolute growth curve of WH; (**c**) relative growth curve of WH.

**Figure 4 animals-13-01545-f004:**
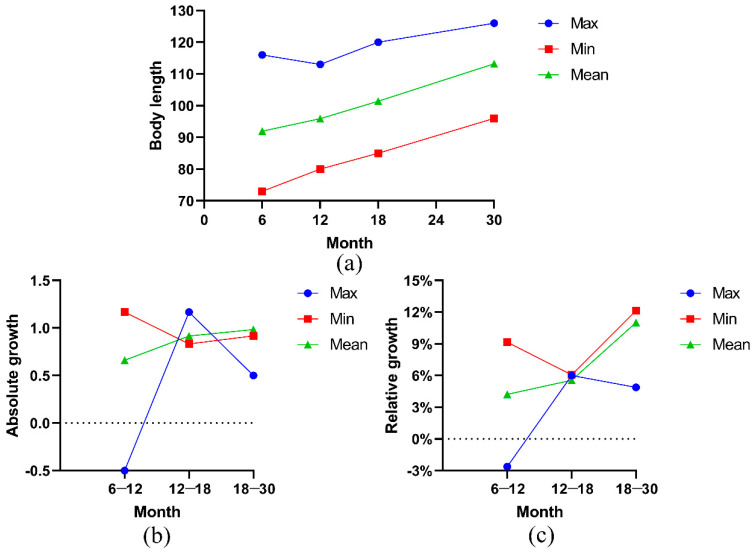
(**a**) Cumulative growth curve of body oblique length (BL); (**b**) absolute growth curve of BL; (**c**) relative growth curve of BL.

**Figure 5 animals-13-01545-f005:**
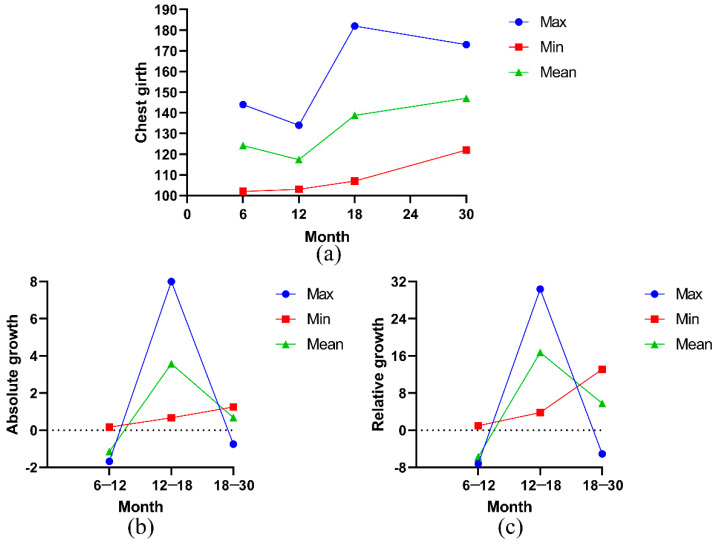
(**a**) Cumulative growth curve of chest girth (CG); (**b**) absolute growth curve of CG; (**c**) relative growth curve of CG.

**Figure 6 animals-13-01545-f006:**
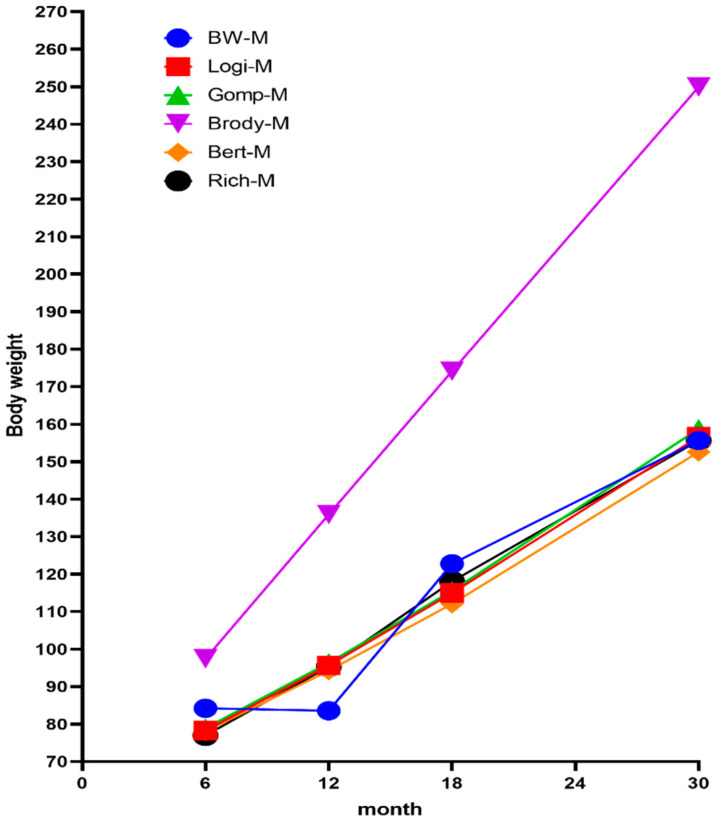
Growth curve of Ashidan yak. BW-M indicated the growth curve of the average weight; Logi-M indicated the growth curve of the Logistic model; Gomp-M indicated the growth curve of the Gompertz model; Brody-M indicated the growth curve of the Brody model; Bert-M indicated the growth curve of the von Bertalanffy model; Rich-M indicated the growth curve of the Richards model.

**Table 1 animals-13-01545-t001:** Growth curve models.

Model	Function	Inflection Point Month, Month	Inflection Point Weight, kg
Logistic	$W=A/(1+Be^{-Kt})$	(LnB)/K	A/2
Gompertz	$W=Ae^{-B\exp(-Kt)}$	(LnB)/K	A/e
Brody	$W=A(1-Be^{-Kt})$	--	--
von Bertalanffy	$W=A{\left(1-Be^{-Kt^{}}\right)}^{3}$	(Ln3B)/K	8A/27
Richards	$W=A/{\left(1+e^{\left(B-Kt\right)}\right)}^{\left(\frac{1}{m}\right)}$	−Ln(m/B)/K	$A/\sqrt[{m}]{m+1}$

Note: W—individual body weight in kg; A—mature body weight in kg; B—time scale parameter; K—growth rate parameter; t—age in months; m—shape parameter; e—natural logarithm.

**Table 2 animals-13-01545-t002:** Descriptive statistics of different growth traits in Ashidan yak.

Month of Age	Trait	Sample Number	Min	Max	Mean	SD	CV
6	BW(kg)	260	58	117	84.22	10.06	11.95%
	WH(cm)	260	81	102	90.71	4.23	4.67%
	BL(cm)	260	73	116	91.96	7.22	7.86%
	CG(cm)	260	102	144	124.22	7.82	6.30%
12	BW(kg)	256	48	113	83.58	10.16	12.16%
	WH(cm)	260	82	108	95.26	4.84	5.08%
	BL(cm)	260	80	113	95.92	5.03	5.24%
	CG(cm)	260	103	134	117.33	4.90	4.17%
18	BW(kg)	226	76	168	122.77	12.92	10.52%
	WH(cm)	251	86	117	99.63	5.45	5.31%
	BL(cm)	251	85	120	101.41	5.61	5.53%
	CG(cm)	251	107	182	138.78	10.49	7.56%
30	BW(kg)	252	108	203	155.63	15.17	9.75%
	WH(cm)	255	90	124	102.53	5.04	5.06%
	BL(cm)	255	96	126	113.23	5.59	4.94%
	CG(cm)	249	122	173	147.03	8.18	5.56%

Abbreviations: BW—body weight; WH—withers height; BL—body oblique length; CG—chest girth; SD—standard deviation; CV—coefficient of variation. Cumulative growth, relative growth and absolute growth of BW of Ashidan yak.

**Table 3 animals-13-01545-t003:** Parameter estimation of different growth curve models of Ashidan yak.

Traits	Models	A	B	K	m
BW	Logistic	300.523	3.763	0.047	
	Gompertz	796.268	2.529	0.015	
	Brody	6527.970	0.991	0.001	
	von Bertalanffy	4065.983	0.750	0.004	
	Richards	155.642	64.941	2.521	70.743
WH	Logistic	104.558	0.154	0.045	
	Gompertz	104.951	0.147	0.041	
	Brody	105.395	0.140	0.037	
	von Bertalanffy	105.092	0.048	0.040	
	Richards	99.848	113.671	6.194	1001.403
BL	Logistic	25,700,000	290,000	0.009	
	Gompertz	1937.138	3.110	0.003	
	Brody	1168.224	0.927	0.001	
	von Bertalanffy	3876.786	0.719	0.001	
	Richards	1410.472	17.499	0.055	6.275
CG	Logistic	14,600,000	128,000	0.009	
	Gompertz	3321.231	3.374	0.003	
	Brody	1679.553	0.933	0.001	
	von Bertalanffy	2533.156	0.645	0.002	
	Richards	2462.751	48.646	0.135	1.836

Abbreviations: A—mature body weight in kg; B—time scale parameter; K—growth rate parameter; t—age in months; e—natural logarithm; BW—body weight; WH—withers height; BL—body oblique length; CG—chest girth.

**Table 4 animals-13-01545-t004:** Goodness of fit of different models for body weight in Ashidan yaks.

Models	MSE	AIC	BIC
Logistic	162.29	4586.50	4600.91
Gompertz	162.99	4590.32	4604.73
Brody	167.49	4614.83	4629.24
von Bertalanffy	163.250	4591.75	4606.16
Richards	154.5	4543.98	4563.19

Abbreviations: MSE—mean square errors; AIC—Akaike’s information criterion; BIC—Bayesian information criterion. The bold indicated the lowest values.

## Data Availability

The study did not report any data.

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
