# Peer review of "Early Growth and Development and Nonlinear Model Fitting Analysis of Ashidan Yak"

_animals, 2023, doi:10.3390/ani13091545_

Round 1
Reviewer 1 Report
Nicely written paper on hornless yak. Here are some suggestions to improve the paper.
* How the hornless yak was developed, gene editing or breeding? Please mention it in manuscript.
* Was 260 sample across all age group enough to draw this conclusion? What is the power of this study?
* Please mention female in the abstract and summary?
* Is this result valid in males too? Discuss about it in the paper?
* No need to mention the excel software in abstract. If mentioned, also include SPSS software too.
Reviewer 2 Report
Meng et al. provided some insight into the growth of the Ashidan Yak. Although the idea and approach are reasonable, the selection of measuring four points during the growth curve is not perfect. The results should the poor fitting of the non-linear models for obtained data. I believe the authors need to collect more samples in order to better draw conclusions about the selected models for the growth curve of this breed.
Line 2: What is “ development law”
Line 14: add “an” before important
Line 14 Might replace “of hornless yaks” with “Ashidan yak”
Line 16: Name four models.
Line 17: it is not clear by “obvious seasonality”
Line 22-23: It is not necessary, instead of this sentence, the authors might describe the data, how many time points, and how the animals were selected.
The authors need to add which models were used, how did the authors compare these models, and why the Logistic model is the best.
The abstract should be rewritten.
Line 35: Please proofread to ensure no double space or other grammar mistakes.
Line 39-40: The authors might add more introduction about how this yak breed was created, and when and which studies have been done in this breed.
Line 73: Why did the authors choose only females?
Line 74: Add more information about the feeding.
Line 75-76: The authors have selected only 4-time points, it is not good enough for fitting the growth curve models.
Line 81: R2 is not suitable for comparing the growth models, the authors should use AIC or BIC and the mean square errors.
Why did the authors choose only three-parameter models, it is known that the four-parameter models such as Richards work better.
Line 98-107: It is not clear; some information might need to refer to table 1.
Line 105-107: More details about the model fitting required
Results: Please define the abbreviation before using them.
Figure 2-5: I am not sure if these results are important, the authors might move them to the supplementary files.
Figure 6: The Figure clearly shows that the authors did not obtain enough time points for fitting the growth models. It is not possible to observe the inflection point/age and the maximum weight and length in the current curve. More data should be used to better fit the models.
Reviewer 3 Report
1. There are some grammatical errors in the manuscript, please correct them carefully.
For example, line 24, “fluctuation trend” ”a fluctuation trend”
“model fitting” “model-fitting”
Line 85, add the space
Line 117, “body weigh” “body weight”
Line 123, add the space
……
2. The reference style should be revised carefully.
3. Figure1 Ashidan yak is not hornless ? The annotation in brackets will make me feel that there are angular types. Please explain. If I 'm not mistaken, I suggest removing the annotation.
4. Please add the reference of the operation model.
5.The “ female Ashidan yaks “ was used in the” Sample collection and growth traits “ section. Please explain why female yak populations are used.
6. Please check the data in the table 3 carefully.
7. line 283 – 292, add some discussion to explain the difference between the previous study and current study.
8. line 295, what’s meaning of “may be related to cow lactation and environment”.
Reviewer 4 Report
Authors are invited to improve on the presentation of their results and ensure that all Figures are properly labelled and appropriately referred to in the text. I have drawn the attention of the authors to some of these errors in the edited draft.
Authors should also ensure that all cited references and appropriately listed and vice versa in line with the guidelines provided by the journal.

Round 2
Reviewer 2 Report
The authors have addressed my comments. Although it is not possible to increase the sample size or add more time points for analyses, the authors have added the Richards model to the analyses. The authors should pay more attention to writing and proofreading.
I have some further suggestions.
Remove some “model” words in lines 16-17, 25-26,
will change with the seasons: might specify more, which kind of changes,
Line 31: did the authors mean mature or maximum body weight?
Line 34: What does it mean “.fivefivefive”. they appeared several times in the manuscript.
Line 35: Too many “growth” in the keywords, suggesting one of key words is non-linear model
Line 63: the program it is not clear which program,
The authors might extend some introduction about the heritability of the growth parameters traits
Pay attention to the font size and formatting,
Line 201: What did the authors mean the “goodness”
Add the discussion of why the Richard model was the best one.
